

# Evaluating outcomes of young forest management on a target species of conservation concern

Henry M. Streby[1], Gunnar R. Kramer[1], Sean M. Peterson[2] and David E. Andersen[3]

[1] Department of Environmental Sciences, University of Toledo, Toledo, OH, United States of America
[2] Department of Environmental Science, Policy and Management, University of California, Berkeley, Berkeley, CA, United States of America
[3] Minnesota Cooperative Fish and Wildlife Research Unit, United States Geological Survey, St. Paul, MN, United States of America

## ABSTRACT

**Background**. Assessing outcomes of habitat management is critical for informing and adapting conservation plans. From 2013–2019, a multi-stage management initiative aims to create >26,000 ha of shrubland and early-successional vegetation to benefit Golden-winged Warblers (*Vermivora chrysoptera*) in managed forested landscapes of the western Great Lakes region. We studied a dense breeding population of Golden-winged Warblers at Rice Lake National Wildlife Refuge (NWR) in Minnesota, USA, where shrubs and young trees were sheared during the winter of 2014–2015 in a single treatment supported in part by the American Bird Conservancy (ABC) and in part by other funding source(s) to benefit Golden-winged Warblers and other species associated with young forest [e.g., American Woodcock (*Scalopax minor*)] and as part of maintenance of early successional forest cover on the refuge.

**Methods**. We monitored abundance of Golden-winged Warblers before (2013–2014) and after (2015–2016) management at the treatment site and a control site, and we estimated full-season productivity (i.e., young recruited into the fall population) on the treatment site from predictive, spatially explicit models, informed by nest and fledgling survival data collected at sites in the western Great Lakes region, including Rice Lake NWR, during 2011 and 2012. Then, using biologically informed models of Golden-winged Warbler response to observed and predicted vegetation succession, we estimated the cumulative change in population recruitment over various scenarios of vegetation succession and demographic response.

**Results**. We observed a 32% decline in abundance of Golden-winged Warbler breeding pairs on the treatment site and estimated a 27% decline in per-pair full-season productivity following management, compared to no change in a nearby control site. In models that ranged from highly optimistic to progressively more realistic scenarios, we estimated a net loss of 72–460 juvenile Golden-winged Warblers produced from the treatment site in the 10–20 years following management. Even if our well-informed and locally validated productivity models produced erroneous estimates and the management resulted in only a temporary reduction in abundance (i.e., no change in productivity), our forecast models still predicted a net loss of 61–260 juvenile Golden-winged Warblers from the treatment site over the same time frame.

Corrected 21 March 2019

Corresponding author
Henry M. Streby,
henrystreby@gmail.com

**Conclusions**. Our study sites represent only a small portion of a large young-forest management initiative directed at Golden-winged Warblers in the western Great Lakes region; however, the brush management, or shearing of shrubs and small trees, that was applied at our study site is a common treatment applied by contractors funded by ABC and its partners on public lands across Minnesota with the expressed intent of benefiting Golden-winged Warblers and related species. Furthermore, the resulting vegetation structure at our treatment site is consistent with that of other areas managed under the initiative, and ABC documents include our study site as successful Golden-winged Warbler management based on observations of ≥1 Golden-winged Warbler at the treatment site since the management. Our assessment demonstrates that, at least for the only site for which pre- and post-management data on Golden-winged Warblers exist, the shearing of shrubs and small trees has had a substantial and likely enduring negative impact on Golden-winged Warblers. We suggest that incorporating region-specific, empirical information about Golden-winged Warbler—habitat relations into habitat management efforts would increase the likelihood of a positive response by Golden-winged Warblers and also suggest that management directed generically at young forest may not benefit Golden-winged Warblers.

## INTRODUCTION

Many species of birds associated with shrubland and early-successional forest cover types are experiencing long-term declines in abundance across eastern North America (*Sauer et al., 2014*). Golden-winged Warblers (*Vermivora chrysoptera*) are a Nearctic-Neotropical migratory species experiencing one such decline (Fig. 1, *Sauer et al., 2014*). Although no cause of the population decline has definitive empirical support, competing hypotheses attribute the declines to breeding-grounds factors such as a reduction in shrubland and young forest area and competition and hybridization with Blue-winged Warblers (*Vermivora cyanoptera*; *Rohrbaugh et al., 2016*), while recent evidence implicates non-breeding factors such as geographic isolation and regional habitat conditions on the wintering grounds (*Kramer et al., 2017*; *Kramer et al., 2018*).

Golden-winged Warblers breed in isolated, high-elevation areas along the Appalachian Mountains and more densely throughout the western Great Lakes region (*Buehler et al., 2007*; *Rosenberg et al., 2016*). Breeding densities of this species generally increase from east to west, with Minnesota hosting nearly half of the global population in 13% of the known breeding distribution (*Rosenberg et al., 2016*). Regional variation in Golden-winged Warbler population declines has likely led to this east-west density gradient, as Appalachian states have experienced declines of >95% since 1965 while Minnesota has seen stable or slightly increasing populations over the same period (Fig. 1; *Zlonis et al., 2013*; *Sauer et al., 2014*; *Rosenberg et al., 2016*). This places a strong stewardship responsibility on the State of

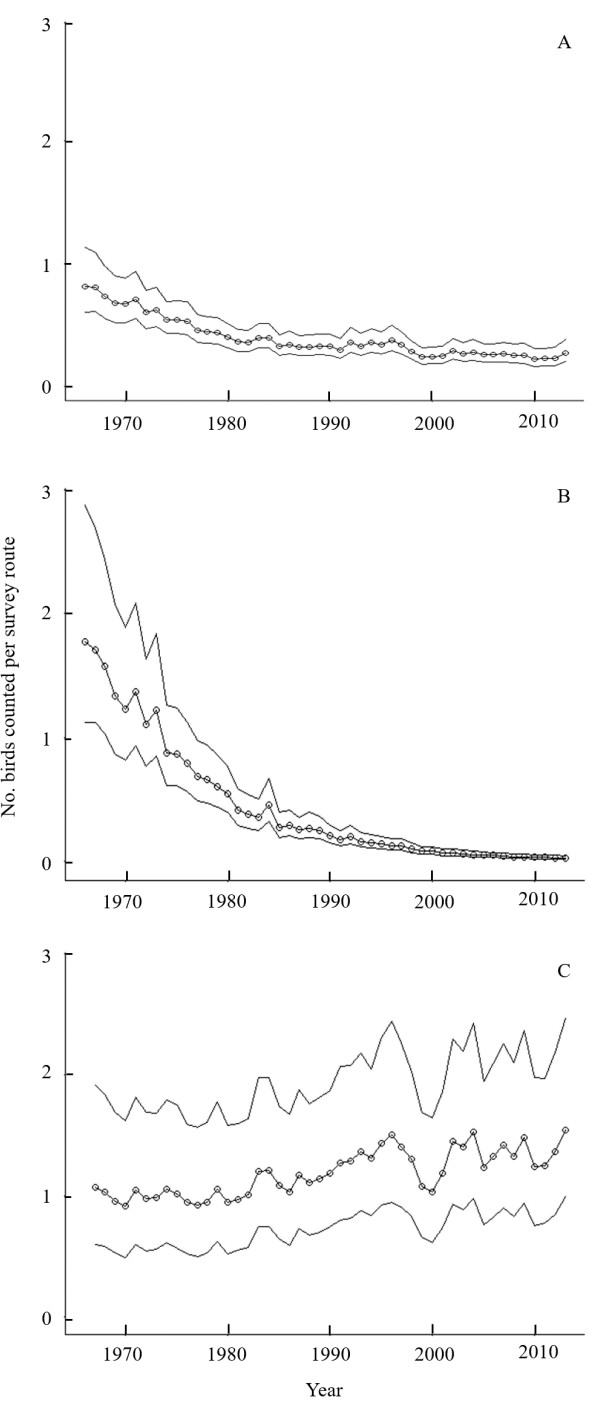

**Figure 1 Long-term population trends for Golden-winged Warblers by region and globally from the Breeding Bird Survey.** Trends in Golden-winged Warbler counts along routes sampled during the Breeding Bird Survey (*Sauer et al., 2014*) from 1966–2013. Modeled numbers of birds per BBS survey route (open circles) and 95% credible intervals are displayed for (A) the global population, (B) the Appalachian breeding distribution segment, which comprises <5% of the global population, and (C) the Minnesota breeding population, which comprises ~50% of the global population. Urgent calls for Golden-winged Warbler conservation action often cite the global trend (A), which has stabilized in recent years because western populations remain stationary or increasing and the influence of the Appalachian population on global trends has decreased as that population asymptotes toward zero.

Minnesota and other jurisdictions in the western Great Lakes region for Golden-winged Warbler conservation as the species is considered for listing under the Federal Endangered Species Act (*Peterson, Streby & Andersen, 2016a*).

In response to long-term declines in Golden-winged Warbler numbers, many studies have been conducted in attempts to inform conservation strategies that might reverse negative population trends (*Buehler et al., 2007*; *Rohrbaugh et al., 2016*). These studies, primarily conducted in the Appalachian Mountains segment of the breeding distribution, have detailed habitat characteristics associated with breeding Golden-winged Warblers from the regional scale (*Crawford et al., 2016*) to the scale of 1-m plots at nest locations (*Aldinger et al., 2015*; *Terhune II et al., 2016*) and many scales in between (*Rossell Jr, Patch & Wilds, 2003*; *Kubel & Yahner, 2008*; *Bulluck & Buehler, 2008*; *Confer, Barnes & Alvey, 2010*; *Aldinger & Wood, 2014*). Although some differences in results may be attributable to differences in methods and variables measured among studies, a generalizable conclusion from the studies that used standardized methods across sites is that breeding habitat structure for Golden-winged Warblers varies among sites, states, and regions, and that site-by-site conservation plans may be required (*Terhune II et al., 2016*; *Streby et al., 2016*).

In 2013 (planned through 2019), multiple large-scale management initiatives began working to manage public and private forested areas to benefit Golden-winged Warblers and other young-forest-associated species in the western Great Lakes region. In July 2013, the American Bird Conservancy (ABC) and partners were awarded $4.7 M to implement management on public and private lands in Minnesota to benefit Golden-winged Warblers (http://www.legacy.leg.mn). In January 2015, ABC announced a $10 M partnership with the US Department of Agriculture's Natural Resource Conservation Service (NRCS) to incentivize and implement the same management prescriptions on private lands in Minnesota and Wisconsin (http://www.abcbirds.org). Together, these initiatives (hereafter; ABC initiative) have the stated goal of managing >26,000 ha within the primarily forested landscape of the western Great Lakes region focused on creating breeding habitat for Golden-winged Warblers (*Johns, Wieber & Otto, 2015*; *Larkin et al., 2016*). ABC predicted this management would "create new breeding habitat for 1,180 breeding pairs of Golden-winged Warblers" and "result in an increase of 16,000 individuals within four years" (*Johns, Wieber & Otto, 2015*, pg 1). The primary focal area of this initiative is the northern hardwood-boreal forest transition zone of northern Minnesota. Forested private and public lands are targeted with financial incentives, and although landscape context is important to songbird conservation (*Donner, Ribic & Probst, 2010*; *Bonnot et al., 2013*), management has been prioritized at accessible sites with secondary consideration of the surrounding mosaic (P Diesser, pers. comm., 2014).

It is critical to monitor effects of focal-species management so that methods can be replicated when successful (*Lyons et al., 2008*) and learned from when not successful (*Hiers et al., 2016*). It is particularly important that conservation actions benefit focal species because policies and plans that are poorly informed and poorly implemented can accelerate the loss of biodiversity (*Woinarski et al., 2017*). The current ABC initiative is occurring in a portion of the Golden-winged Warbler breeding distribution that supports the bulk of breeding Golden-winged Warblers and lessons from this initiative
are potentially beneficial for informing Golden-winged Warbler management where their populations are in peril. This initiative's management is guided by regional Best Management Practices (BMPs) for Golden-winged Warblers (*Larkin et al., 2016*) and based on a Golden-winged Warbler conservation plan (*Roth et al., 2012*). Although Minnesota hosts approximately half of the global breeding population of Golden-winged Warblers, >95% of the data used to inform the *Roth et al. (2012)* plan were collected in the nearly extirpated Appalachian Mountains breeding population segment. Therefore, timely assessment of the current management initiative is necessary both because of the immediate need for model management strategies for Golden-winged Warblers elsewhere and because of the potential for necessary adjustments during implementation due to the plan's reliance on Golden-winged Warbler—habitat relations derived from other portions of the breeding distribution.

As part of an ongoing study of Golden-winged Warbler—habitat relations in the western Great Lakes region, we had the fortuitous opportunity to study the response of Golden-winged Warblers to management intended to create young forest conditions thought to favor Golden-winged Warblers and other species associated with young forests at Rice Lake National Wildlife Refuge (NWR) in east-central Minnesota. Our study is the only assessment of habitat management, at least in part intended to benefit Golden-winged Warblers, with extensive before- and after-management information on habitat use and population dynamics, and therefore may provide insight into the potential response of Golden-winged Warblers to management intended to increase population size and productivity.

We monitored abundance of breeding pairs of Golden-winged Warblers in a managed area within Rice Lake NWR for 42 years prior to management and two years after management, and in a nearby area that did not experience young-forest management, in a before-after-control-impact study design. In addition, because abundance alone can be a misleading indicator of habitat quality (*Van Horne, 1983*), we used biologically informed, spatially explicit models of full-season productivity (i.e., young recruited into the fall population; *Peterson, Streby & Andersen, 2016a*) of Golden-winged Warblers before and after management to estimate changes in productivity in response to management. Our study represents only two sites (one treatment and one control) within multiple large-scale young-forest management efforts in the region, and we present our assessment with uncertainty about its application to broader initiative outcomes. However, the management at our study site is described by the implementers as identical to that applied elsewhere in Minnesota and is reported as a successful management site for Golden-winged Warblers (*Larkin et al., 2016*), so it is reasonable to conclude that the outcomes at our site are not unique.

## MATERIALS AND METHODS

### Study area

We studied Golden-winged Warblers at two sites within Rice Lake NWR (46.528179°N, −93.407202°W) in Aitkin County, Minnesota, USA. Rice Lake NWR encompasses 7,300

ha of diverse cover types including lakes, rivers, grassy and shrubby wetlands, bogs, upland and wetland forests of various successional stages, and minimal agriculture. The refuge hosts areas with some of the highest known density of breeding Golden-winged Warblers, including locations with >1 breeding pair/ha. Prior to this management action, our 80-ha main (treatment) study site included 41 ha of shrubland and early-successional forest cover types and supported 62 breeding pairs of Golden-winged Warblers, and our 30-ha control site included 16 ha of shrubland and early-successional forest cover types and supported 19 breeding pairs of Golden-winged Warblers. Our treatment site and control site were separated by 2.0 km and some of the area between sites was also occupied by Golden-winged Warblers. Both study sites combined supported one pair per 0.7 ha of shrubby cover types with which the species is typically associated as a habitat specialist (*Ficken & Ficken, 1968*; *Confer, Hartman & Roth, 2011*), and we are unaware of any more densely populated areas within the Golden-winged Warbler breeding distribution.

Prior to this management action, vegetation in both our treatment and control sites was a diverse and patchy mix of trees, shrubs, sedges, and forbs with soft, feathered-edge (i.e., small-scale complexity) transitions between shrubby areas and later-successional forest (Fig. 2). The vegetative structure of the shrublands in our study area was naturally maintained by poor soils, and no management had occurred on our treatment site in at least the previous 13 years (W Ford, 2011, pers. comm. and H Streby, 2017, personal observation using Google Earth ®). Common trees in later-successional forest stands included sugar maple (*Acer saccharum*), red maple (*A. rubrum*), quaking aspen (*Populus tremuloides*), bigtooth aspen (*P. grandidentata*), paper birch (*Betula papyrifera*), American basswood (*Tilia americana*), bur oak (*Quercus macrocarpa*), and red oak (*Q. rubra*). Shrubland and young forest stands were a complex mix of willow (*Salix* spp.), hazel (*Corylus* spp.), dogwood (*Cornus* spp.), paper birch, quaking and bigtooth aspen, grasses, sedges, and forbs with individual and small groups of young and mature bur oak and red oak trees. The complexity and diversity of vegetation was reflective of the diversity of soil types including hamre muck, Dysler silt loam, and talmoon fine sandy loam (*NRCS Soil Survey Staff, 2016*). Due to the dense, ephemerally wet, poorly drained nature of these soils, the trees and shrubs of the managed area were slow growing, and there had been no discernable change in vegetation structure or composition during the four years prior to treatment (The Authors, personal observation) or for five years prior to our observations (A Hewitt, pers. comm., 2014).

## Management

During the winter of 2014–2015, several areas were managed in a single harvest prescription across Rice Lake NWR. This prescription included ~40 ha of vegetation shearing of which 12 ha were in our treatment study site and none were within our control site. Management planning documents included maps with the area to be managed in our treatment study site described as "GWWA thinning" (Supplemental Information 1). The managed areas were mechanically cleared of vegetation to ground level through hydro-axing, or shearing, during the winter when frozen ground enabled accessibility, as per recommendations in the Golden-winged Warbler Conservation Plan (*Roth et al., 2012*). This management

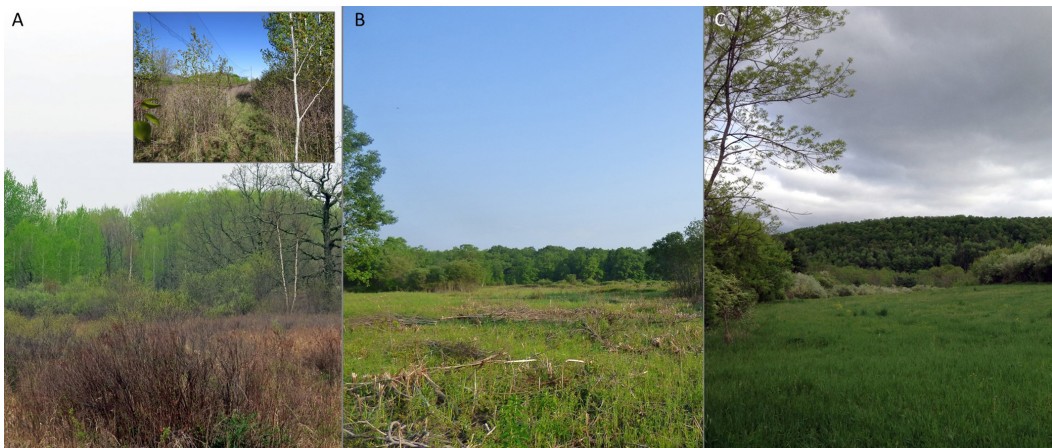

**Figure 2** **The study site before and after management and a comparable area managed under the same prescription with the same result.** Vegetation at Rice Lake National Wildlife Refuge in Minnesota (A) before and (B) during the first growing season after management intended to benefit Golden-winged Warblers and associated young forest species and (C) a landscape described as managed for Golden-winged Warblers in Bald Eagle State Park in Pennsylvania (*Bakermans, Ziegler & Larkin, 2015*, J Larkin, ABC, pers. comm., 2015). Inset in A demonstrates the complexity in pre-management vegetation structure at our treatment study site. Image C demonstrates that management in Minnesota replicated vegetation structure of landscapes described as managed areas for Golden-winged Warblers by an ABC employee in the Appalachian region. Photos by H Streby (A and inset) and G Kramer (B and C).

prescription has been described as the clearing of overgrown brush (*Dieser, 2017*), and is the primary technique implemented on public lands in Minnesota during the ABC initiative (*Larkin et al., 2016*; *Dieser, 2017*).

Sometime after the management was completed, employees of the ABC and Rice Lake NWR observed the resulting vegetation structure and determined that some sheared areas (primarily those outside of our treatment site) were consistent with Golden-winged Warbler BMPs, and ABC provided funding for those portions of the management (S Graff, ABC, pers. comm., 2018). Those employees determined that other sheared areas (primarily those within our treatment site) were more consistent with American Woodcock (*Scolopax minor*) BMPs, and those portions of the management were funded by other sources. The management we assessed was described as Golden-winged Warbler management prior to the management implementation; the management was applied as a single consistent prescription across our treatment site; and the management technique was shearing, which is the method commonly used by ABC and its partners in an initiative aimed at benefiting Golden-winged Warblers across Minnesota.

Following management, the resulting vegetation structure in the 2015 growing season was dominated by patchy areas of sedges, forbs, bare ground, and woody debris (Fig. 2), similar to previous management by this method in Minnesota (e.g., *Hanowski, Christian & Nelson, 1999*). The pre-management soft, feathered transition into the surrounding later-successional forest was sheared, resulting in a hard, linear edge between the managed area and the surrounding forest. In 2016, the shorn stumps of trees and shrubs began to re-sprout stem clusters creating low (<0.25 m), dense, homogenous areas of woody

stems within a mosaic of sedges and forbs (D Andersen & G Kramer, pers. obs., 2016). In 2017, the managed area was dominated by a low (0.25–0.50 m), dense mat of dogwood, typical of other shrublands managed by shearing in Minnesota (e.g., *Hanowski, Christian & Nelson, 1999*). The stated goal of the vegetation management was to create shrubby young forest with high complexity that would benefit Golden-winged Warblers and other young forest-associated species within four years (*Larkin et al., 2016*). After three growing seasons, the regenerating managed area was visually similar to a nearby ephemeral wetland that was dominated by sedges and low shrubs and has not hosted Golden-winged Warblers during any of our observations. Consistent with management by the same prescription previously in Minnesota (*Hanowski, Christian & Nelson, 1999*), after three years vegetation on the treatment area was behind pace to recreate the vegetation structure present prior to management and that described in the Golden-winged Warbler BMPs (*Roth et al., 2012*).

## Golden-winged Warbler abundance monitoring

From 2013–2016 we censused Golden-winged Warbler breeding pair abundance in the treatment and control sites as part of two studies that involved individual marking and monitoring of males and females (*Peterson et al., 2015*; *Kramer et al., 2018*). In each year, we conducted singing male surveys of our treatment site and control site, and we mist-netted and color banded males and females, and spot-mapped core breeding territories for males. Spot-mapping is not adequate for delineating complete territories or home ranges of Golden-winged Warblers, but it is useful for identifying and counting unique individual males in an area (*Streby, Loegering & Andersen, 2012*). Spot-mapping is also not adequate for determining pairing success or mating success; we therefore employed additional monitoring techniques, including behavioral observations, nest searching, netting, and individually marking females, to confirm that all of the males in our study area were paired with females that were initiating nesting attempts. In all 4 years, we believe our methods resulted in a complete count of Golden-winged Warbler pairs within our study sites. Due to logistical constraints, we did not assess Golden-winged Warbler abundance at these study sites in 2017, but only observed vegetation structure.

## Productivity modeling

Data we used to inform Golden-winged Warbler productivity models were collected under protocols #1004A80575 and #160333530A approved by the University of Minnesota Institutional Animal Care and Use Committee, and research permits #21631 (USGS Bird Banding Laboratory) and 19017 (Minnesota Department of Natural Resources) during studies of nesting and post-fledging ecology of Golden-winged Warblers in 2011 and 2012. We applied spatially explicit models of full-season productivity (*Peterson, 2014*; *Peterson, Streby & Andersen, 2016a*) for Golden-winged Warblers and estimated the number of young recruited into the fall population under multiple scenarios. Our models were informed by data about Golden-winged Warbler nest productivity ($n = 216$ nests) and fledgling survival ($n = 190$ fledglings) collected during a two-year study at three sites in northern Minnesota and southeastern Manitoba, Canada, including our study area at Rice Lake NWR (*Peterson, Streby & Andersen, 2016a*). For field data collection details, see *Peterson (2014)*

and *Peterson, Streby & Andersen (2016a)*. Briefly, we used radio-telemetry monitoring of breeding females and standard nest-searching methods to locate Golden-winged Warbler nests and monitor their fates. We radio-marked nestlings and used their fates before and after fledging to determine nest success, fledged brood sizes, and fledgling survival rates during and after dependence on adult care (*Streby et al., 2014*; *Streby et al., 2015*). We assessed the effects of micro- and meso- scale habitat characteristics on nest and fledgling survival rates and found stand-level effects (i.e., the identity and area of cover types around a nest) to be significantly more influential on survival rates in both stages (*Streby et al., 2014*), and we therefore informed our full-season productivity models with these larger-scale parameters as in *Peterson (2014)* and *Peterson, Streby & Andersen (2016a)*.

Step-by-step instructions and programming code for parameterization and implementation of full-season productivity models are detailed in Appendix C of *Peterson (2014)* and further described in *Peterson, Streby & Andersen (2016a)*. Summarily, full-season productivity models are logistic exposure survival models (*Shaffer, 2004*) that incorporate effects of landscape composition and configuration at statistically and biologically relevant scales on nest success and fledgling survival rates. In these models, the relationship between each cover type and predicted nest success or fledgling survival rates varies depending on the presence and area of that cover type and other cover types at predetermined, biologically relevant, radii from any given location on the landscape. When combined, estimates of nest success, fledged brood size, renesting rates, and fledgling survival rate for every square meter in the landscape result in a spatially explicit map of full-season productivity for Golden-winged Warblers in an area of interest. Specifically, full-season productivity models produce an estimate of the number of young recruited into the fall migratory population per breeding pair in the modeled area. Multiplying the full-season productivity estimate by the number of breeding pairs in the modeled area produces an estimate of the number of young recruited into the fall population from the population of birds breeding in the area of interest. By changing the shape, area, or cover type identity of any stand in the modeled landscape, and then re-applying the model to the resulting landscape, one can estimate full-season productivity for any hypothetical or real stand-level management scenario. These models have been validated as meaningfully predictive for Golden-winged Warbler productivity data in our study area (*Peterson, Streby & Andersen, 2016a*) and we therefore expect their predictions of post-management productivity to provide insight into how Golden-winged Warblers respond to changes in cover-type composition resulting from management. Full-season productivity models were developed as an improvement upon models including only nesting success because fledgling survival rate is an increasingly appreciated component of songbird productivity (*Streby et al., 2014*), and fledgling survival rate and nesting success can be influenced differently by the same habitat characteristics (*Cohen & Lindell, 2004*; *Jackson, Froneberger & Cristol, 2013*; *Streby et al., 2014*), making the inclusion of fledgling survival rate and habitat use critical management considerations (*Peterson, Streby & Andersen, 2016a*; *Peterson, Streby & Andersen, 2016b*).

We estimated Golden-winged Warbler full-season productivity for the treatment site at Rice Lake NWR under three scenarios of forest succession: (1) rapid succession, (2)

moderate succession, and (3) slow succession. In each model scenario, we assumed that shrub and small tree density would regenerate to a state similar to pre-management vegetation within four years (rapid; as assumed by the ABC initiative), 10 years (moderate; possible based on historical succession), and 20 years (slow; likely based on historical succession). In addition, we estimated full-season productivity in the absence of management (i.e., as if management had not occurred) to provide an estimate to which we could compare the net effects of the range of post-management scenarios. In the no-management scenario, we assumed the area would maintain similar, pre-management, vegetation structure for the entire period, which is likely for all three scenarios given the soil types and vegetation succession history of the managed area. It is possible that the complexity and vertical stratification of the pre-treatment vegetation structure may not return in any of the post-management scenarios, but we assumed that adequate shrub and small tree cover would regenerate by the end of each scenario to host pre-treatment numbers of breeding Golden-winged Warblers. For each succession scenario we modeled the response of Golden-winged Warbler productivity two ways. First, we assumed Golden-winged Warbler productivity responded linearly with time in each scenario, which is likely given the relationship between Golden-winged Warbler productivity and landscape composition and the slow and approximately linear vegetation succession that occurs in our study area. Second, although there is no evidence of density dependent changes in productivity in our study system, population theory predicts that productivity should increase in response to reduced density. Therefore, in a second set of scenarios, we assumed that after a brief initial negative response, Golden-winged Warbler productivity would increase rapidly to a level 25% greater than pre-management productivity, followed by a return to pre-management productivity by the end of each scenario period. Each of the various productivity scenarios included a linear response in density as described above. We did not consider models in which density responded in a parabolic fashion because our study sites hosted the greatest known densities of breeding Golden-winged Warblers prior to management, and any increase in density is unlikely.

We assessed only abundance under post-management conditions for our control site (i.e., no productivity models applied) because we did not expect vegetation-based changes in productivity in the absence of management within any of our scenario time frames. Certainly, trees and shrubs will grow larger on the control site and in our no-management scenario, but past succession rates indicate it is unlikely that any shrub-dominated stands in our study area would succeed to forest-dominated stands in the time frame of our models.

## RESULTS

During the pre-management breeding seasons of 2013–2014, our treatment study site hosted 62 pairs of breeding Golden-winged Warblers each year (Fig. 3) and our control site hosted 19 breeding pairs each year. In 2015, the breeding season immediately following management, our main site hosted 45 breeding pairs of Golden-winged Warblers (Fig. 3) and the control site hosted 19 pairs. One male that bred in the treatment site for two years prior to management, established a territory 2.6 km away in the control site

following management, apparently displacing or replacing the male that previously bred in that control-site territory the previous year. In 2016, the second breeding season after management, our treatment site again hosted 45 breeding pairs of Golden-winged Warblers (Fig. 3) and the control site again hosted 19 pairs. Within the treatment site, primary losses of breeding pairs occurred in the middle of managed areas, and pairs remaining along the edges of managed areas shifted core use areas away from managed vegetation and into adjacent later-successional forest (Fig. 3).

For all post-management scenarios, we included the observed 45 breeding pairs for 2015, 45 breeding pairs for 2016, and then included linear responses from 45 back to 62 breeding pairs in the remainder of the scenario period. For example, in the rapid succession scenario, 2015 and 2016 were included as the first two years after management, and our models assumed 53 breeding pairs in 2017 and 62 in 2018, to complete the return to pre-treatment abundance in four years after management. We did the same for the moderate and slow succession scenarios including the first two years as observed and then return from 45 to 62 breeding pairs linearly over the subsequent eight years and 18 years to complete the scenario periods, respectively.

Prior to management, our treatment site produced an estimated 1.45 (±0.11) juveniles recruited into the fall population per breeding pair each year (Fig. 4). Given the abundance of 62 breeding pairs, we estimated that 90 (±6) juvenile Golden-winged Warblers were recruited into the fall population each year from this site prior to management. After management, the treatment site produced an estimated 1.04 (±0.10) juveniles recruited into the fall population per breeding pair per year (Fig. 4). In our linear productivity models, we assumed productivity would return to pre-management levels linearly with time and reach 1.45 fledglings per pair by the end of each scenario period. Given the abundance of 45 breeding pairs in 2015 and the modeled 1.04 (±0.10) juveniles per breeding pair, we estimated that 47 (±4) juvenile Golden-winged Warblers were recruited into the fall population from our treatment site in 2015, and the number of juveniles recruited into the fall population increased as abundance and productivity increased linearly with time. In our density-dependent productivity models, we used the same initial post-management abundance observations and productivity estimates, but then assumed productivity increased relatively rapidly, peaking at 1.81 juveniles per breeding pair (i.e., 25% higher than pre-management productivity) and then returned to the pre-management 1.45 juveniles per breeding pair by the end of each scenario (see Supplemental Information).

Under the rapid succession scenario, we estimated that 262 (±19) juvenile Golden-winged Warblers were recruited into the fall population from the treatment site during the 4 years following management, compared to 360 (±22) juveniles during the same period if management had not occurred. Under the moderate succession scenarios, our linear productivity model estimated that 662 (±48) juvenile Golden-winged Warblers were recruited into the fall population from the treatment site during the 10 years following management, compared to 825 (±59) juveniles estimated from the density-dependent, parabolic productivity model and 900 (±64) juveniles during the same period if management had not occurred. Under the slow succession scenario, we estimated that 1,340 (±96) juvenile Golden-winged Warblers were recruited into the fall population from

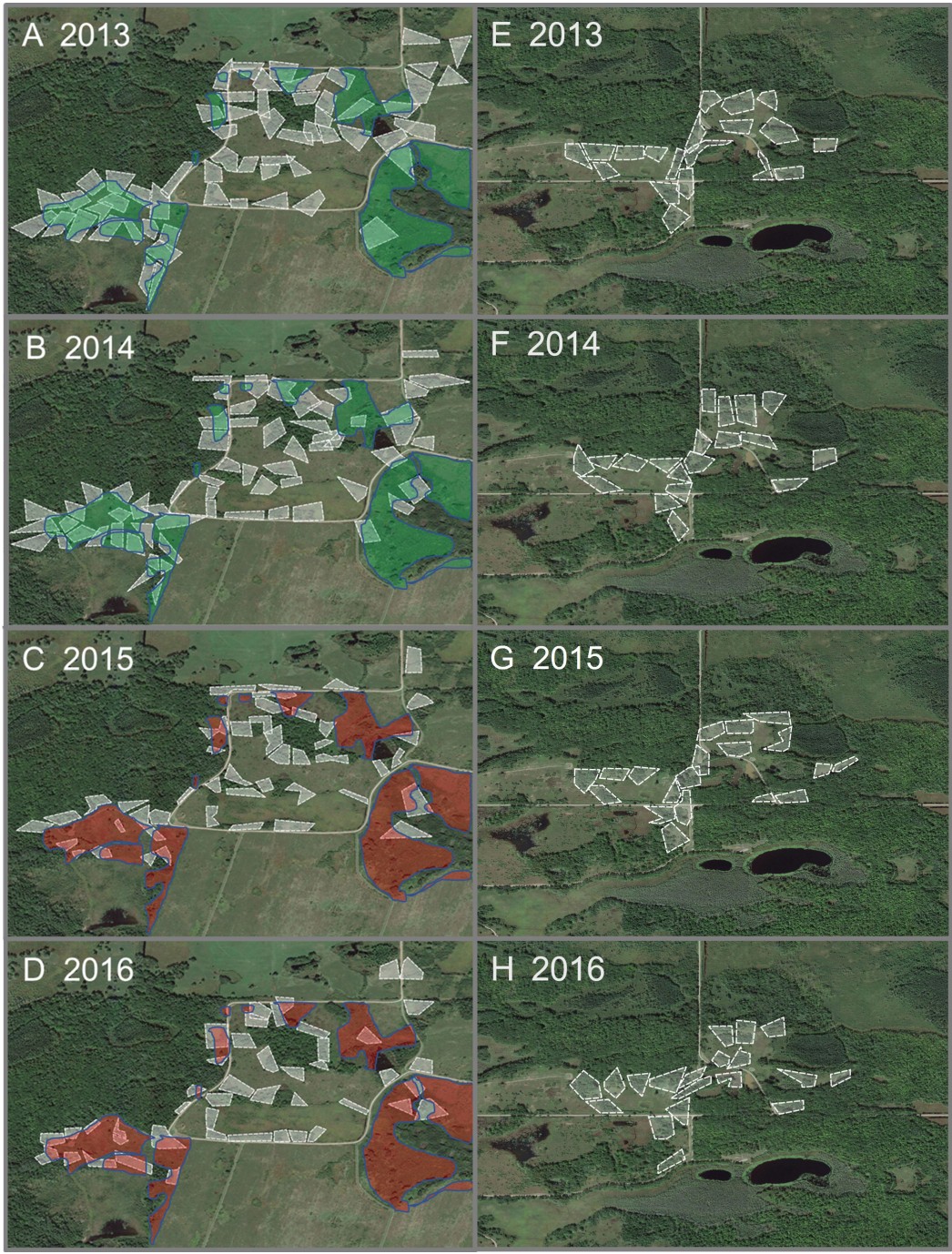

**Figure 3 Core territories of breeding pairs of Golden-winged Warblers before and after management.**
Core breeding territories (white polygons) of Golden-winged Warbler pairs in our study sites at Rice Lake National Wildlife Refuge during 2013–2016. Core breeding territories in the treatment site are displayed for (A) 2013 ($n = 62$ pairs) and (B) 2014 ($n = 62$ pairs) before vegetation shearing, and (C) 2015 ($n = 45$ pairs) and (D) 2016 ($n = 45$ pairs), the 2 breeding seasons following 

**Figure 3 (…continued)**
vegetation shearing. Areas of vegetation shearing are identified by green polygons before management and red polygons after management. Core breeding territories in the control site are displayed for the same 4 years (E–F) with 19 pairs in each year. All core breeding territories were delineated based on spot-mapping, mist-netting, and observations of nest building, nestling feeding, and territorial behavior, are intended for census information, and do not represent total area used by birds for their song territories or home ranges (*Streby, Loegering & Andersen, 2012*). Base map from ESRI World Imagery.

the treatment site during the 20 years following management, compared to 1,674 ($\pm$120) juveniles estimated from the density-dependent, parabolic productivity model and 1,800 ($\pm$129) juveniles during the same period if management had not occurred. Therefore, in the rapid, moderate, and slow succession scenarios we estimated that 12 ha of shearing resulted in a net loss of 98, 238, and 460 juvenile Golden-winged Warblers under our linear productivity models, respectively, at the treatment site. Our density-dependent, parabolic productivity models estimated a net loss of 72 juveniles over 10 years or 96 juveniles over 20 years, compared to recruitment under our no-management scenario.

Although our productivity models were informed by a sizeable, locally relevant, dataset and have been validated as predictive of Golden-winged Warbler productivity in our study area (*Peterson, Streby & Andersen, 2016a*), we do not have direct measures of post-management productivity to confirm model accuracy. However, even in a scenario under which only abundance, and not per-pair productivity, was influenced by the management, our forecast models predicted a net loss of 61, 137, and 260 juvenile Golden-winged Warblers recruited into the managed population in the rapid, moderate, and slow succession scenarios, respectively. There is no biologically plausible recruitment rate that this population could reach within 10–20 years after management that would compensate for the immediate decline and expected slow recovery in breeding abundance.

# DISCUSSION

We observed an immediate decline in breeding Golden-winged Warbler abundance following management consisting of shearing dense and diversely stratified vegetation surrounded by later-successional forest at Rice Lake NWR, while abundance remained unchanged at a nearby control site. Our observations are consistent with *Hanowski, Christian & Nelson (1999)* who reported that mechanical clearing of Minnesota shrublands had negative and persistent effects on occupancy and abundance of shrubland and edge-associated birds including Golden-winged Warblers. We did not include detailed measurements and comparisons of vegetation structure before and after management because (1) detailed vegetation structure descriptions of our study site are published (e.g., *Peterson, Streby & Andersen, 2016a*), (2) Golden-winged Warbler productivity and survival relate to cover-type patch-scale habitat characteristics more strongly than to microhabitat vegetation structure (*Streby et al., 2014*), and (3) statistics are unnecessary for comparing the sizes of trees and shrubs to sedges and forbs (*Johnson, 1999*).

Spatially explicit models of Golden-winged Warbler full-season productivity indicated a decline in per-pair productivity concurrent with the decline in abundance after management. The initial decline in productivity was associated primarily with increased

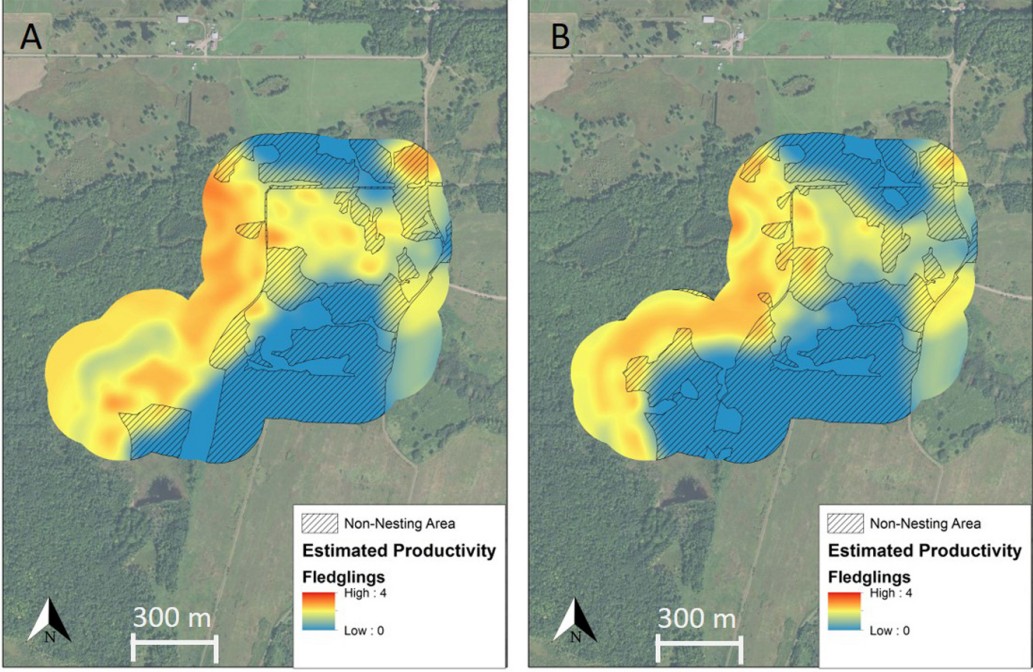

**Figure 4** **Spatially explicit model of full-season productivity for Golden-winged Warblers breeding in the study area before and after management.** Productivity surfaces derived from spatially explicit full-season productivity models for Golden-winged Warblers breeding at our treatment site at Rice Lake National Wildlife Refuge, Minnesota, USA (A) before and (B) after management implemented with the intention of benefiting the species and related young forest-associated species. Non-nesting area (hatched) is comprised of roads, gravel parking areas, and grassland and productivity values from non-nesting areas are not included in total productivity estimates. Management increased the amount of non-nesting area and decreased productivity for the remaining nesting area from 1.45 fledglings raised to independence per breeding pair prior to management to 1.04 fledglings per pair immediately after management. Base map from ESRI World Imagery (both images are identical for ease of comparison of model outputs, thus the base maps are not intended to display location of vegetation shearing).

area of grassland on the landscape, which has negative effects on Golden-winged Warbler nest productivity and fledgling survival rate in most landscape configurations in Minnesota (*Peterson, 2014*) despite being identified as an important habitat component at the micro-habitat scale in the Appalachian breeding distribution segment (*Terhune II et al., 2016*). Our model of an unlikely, 4-year recovery of the vegetation and return of Golden-winged Warbler abundance and productivity to pre-management levels, based on the stated timeline from ABC and its partners, estimated a 27% reduction in juveniles recruited into this population. Our models of more likely scenarios of vegetation succession and Golden-winged Warbler response over 10 and 20 years after management estimated that this population will produce 26% fewer juveniles over those periods than if the management had not occurred. These losses equate to 20–38 Golden-winged Warblers lost per hectare of vegetation shearing at our study site. Even in density-dependent, parabolic productivity models, in which productivity rapidly increased to 25% greater than pre-management levels and then returned to pre-management levels by the end of each scenario, we still

estimated a net loss of 5–8% in juveniles recruited over 10–20 years post-management, primarily because no realistic level of productivity could compensate for the substantial decline in abundance. As a point of emphasis, our productivity estimates refer to the number of young recruited into the fall migrating population and our models already include the large amount of mortality that occurs during the post-fledging period; had we estimated productivity by the more traditional measure of number of young fledged from nests, the negative effect on productivity would appear greater in all scenarios.

It is unclear to us why large-scale vegetation management targeting Golden-winged Warblers was initiated in Minnesota in general and why management for young forest cover types would occur at a location with the highest recorded density of breeding Golden-winged Warblers at Rice Lake NWR in particular. During structured decision making in management and conservation planning a critical initial step is to determine if action is necessary (*Lyons et al., 2008*); the dense, highly productive, and stable or increasing population of Golden-winged Warblers in Minnesota seems not to meet this criterion, and there was not a denser Golden-winged Warbler breeding population known than the one at our study site prior to management. We speculate that this initiative may have been targeted in Minnesota in response to dramatized reports of the global decline of Golden-winged Warblers without consideration of the great regional variation in population trends including the long-term stability of the Minnesota population (e.g., *Cornell Lab of Ornithology, 2016*), and by erroneous descriptions of the Minnesota population being in decline (e.g., *MacSwain, 2016*).

If management targeting Golden-winged Warblers was warranted in Minnesota, creating patches of diverse vegetation structure within otherwise contiguous mature forest areas would likely result in increased population size, based on extensive research on the species in Minnesota (*Peterson, Streby & Andersen, 2016a*; *Peterson, Streby & Andersen, 2016b*; *Streby et al., 2016*). However, Minnesota already surpasses the percent forested area in early-successional stages (i.e., 17%) recommended by the conservation plan (*Roth et al., 2012*) on which the ABC initiative management is based and ongoing forest management in Minnesota has maintained or increased that percentage over at least the past three decades (*Miles & VanderSchaaf, 2012*). Shearing vegetation that currently supports dense and productive populations of Golden-winged Warblers is unlikely to increase their abundance and resulted in loss of productivity in all of our projections. Additionally, increases in grassland area, even temporarily, potentially also compromise Minnesota's goals of managing and maintaining various successional stages of forest (*Johnson et al., 2008*) that provide resources for many species of birds during the nesting (*Niemi et al., 2016*) and post-fledging periods (*Streby et al., 2011*) in a region that has already experienced long-term losses in forest functional and structural heterogeneity (*Schulte et al., 2007*).

Although Minnesota hosts approximately half of the global breeding population of Golden-winged Warblers, only 4% (18 out of 412 nesting territories) of the data used to inform the *Roth et al. (2012)* conservation plan were collected in Minnesota and no information regarding post-fledging habitat requirements was included in that plan. Studies on Golden-winged Warblers in Minnesota have demonstrated the importance of later-successional forest during the nesting season (*Streby, Loegering & Andersen, 2012*),

the dependent post-fledging period (*Peterson, 2014*; *Streby et al., 2014*; *Peterson, Streby & Andersen, 2016a*), and for fledglings after independence from adult care (*Streby et al., 2015*). Given the lack of information derived from this core area of Golden-winged Warbler breeding density in the *Roth et al. (2012)* conservation plan, it is not clear why the Upper Mississippi River and Great Lakes Joint Venture concluded that "Golden-winged Warbler needs are met" for the region by the *Roth et al. (2012)* conservation plan (http://www.uppermissgreatlakesjv.org/Priorities). Even in areas where the *Roth et al. (2012)* conservation plan includes locally relevant data, such as the Pocono Mountains of eastern Pennsylvania, Golden-winged Warblers continue to decline despite large-scale management actions (*Fearer, 2016*). These continued declines in the Appalachian segment of the species' breeding distribution suggest that management is not addressing the factors driving population declines or that the limiting factors for Golden-winged Warbler population growth exist outside the breeding season, or both. Recent information (*Kramer et al., 2017*; *Kramer et al., 2018*) suggests that declines in Golden-winged Warbler abundance may be driven by factors outside of the breeding distribution for the Appalachian Mountains breeding distribution segment.

Our study occurred at only a single treatment and control site in one study area, and it is possible that the vegetation management at our study area, or the response of Golden-winged Warblers at our study area, was unique and not representative of the response of Golden-winged Warblers to vegetation management implemented as part of the broader ABC-led Golden-winged Warbler initiative in the region. We also recognize that management decisions at the scale of a local refuge can be influenced by factors other than maximizing the benefit to a single species of conservation concern. However, a post-management report from individuals involved with the ABC initiative includes our treatment site as one of many sites where "shrub management" was used to "create habitat for Golden-winged Warblers" (*Larkin et al., 2016*, pg. 3,18), and shrub management was the primary method used on public lands during this initiative (*Dieser, 2017*). This indicates that those who supported the management and are conducting post-management surveys identify the management at our treatment site as standard, and furthermore describe it as successful with respect to Golden-winged Warblers (*Larkin et al., 2016*), regardless of post-management determinations of species expected to benefit from particular portions of the management (see 'Management' section).

It is not clear whether results like ours could have been identified if management had not occurred on a study site where Golden-winged Warblers were already intensively studied, as the protocols followed by ABC and their partners in Minnesota did not include pre-management bird surveys. The post-management surveys being conducted as part of the ABC initiative are designed only to detect the presence of Golden-winged Warblers and other species purported to benefit from the management (*Larkin et al., 2016*). Without pre-management data and data from control sites, there is little insight to be gained about the effects of management based solely on post-management surveys for species presence. In this instance, Golden-winged Warblers persisted at our study area following management, and post-management assessment could erroneously conclude that they responded positively to management based solely on presence following management.

Although our spatially explicit models of Golden-winged Warbler full-season productivity are locally and biologically informed and have been validated in our study region to be predictive of observed productivity (*Peterson, Streby & Andersen, 2016a*; *Peterson, Streby & Andersen, 2016b*), direct studies of full-season productivity at a sample of managed sites would be necessary to determine the full effect of the Golden-winged Warbler management initiative in Minnesota and the broader western Great Lakes region. To our knowledge, there are no current plans to fund or conduct such an assessment. The long-term stability and strong productivity of Minnesota's Golden-winged Warbler population and the results of our monitoring and analysis suggest that the management initiative in Minnesota may not be meeting its stated goals and is, at least at one representative site, producing conditions that result in negative impacts on the target species.

## CONCLUSION

We suggest that there are several important implications from our assessment. First, we think this highlights the possible negative consequences of basing management prescriptions on plans developed in ecologically and geographically disparate systems. For example, it is possible that mechanical clearing of shrublands, as applied at our site and as applied by ABC and its partners across Minnesota, produces the initiative's intended outcome in other regions, but is known not to do so in Minnesota (e.g., *Hanowski, Christian & Nelson, 1999*). In addition, not incorporating appropriate assessment efforts, and determining which species were targeted by management only after the management has been completed, may obfuscate the effects of management, and worse, may result in incorrect conclusions about management effects on pre-management target populations. For example, the *Larkin et al. (2016)* report includes our treatment site as successful management because they detected ≥1 Golden-winged Warbler during surveys following the management. However, the *Larkin et al. (2016)* report includes only sites that were managed following BMPs for Golden-winged Warblers, and it is therefore unclear if those surveys include the portions of management at Rice Lake NWR that were identified as consistent with American Woodcock BMPs after the management was completed despite being part of the same treatment. Furthermore, not incorporating local and regional information about factors that influence succession (e.g., local soil types, vegetation growth rates, etc.) into management plans may increase the period over which management activities affect target populations, and in this case, extend the period of reduced Golden-winged Warbler abundance and productivity. Although attractive to planners and managers, overly precise management prescriptions that fail to incorporate local circumstances and diverse conservation needs can lead to expensive and expansive conservation failures (*Hiers et al., 2016*) and even extinctions (*Woinarski et al., 2017*). For Golden-winged Warblers in the western Great Lakes region, we suggest that conservation plans based on information derived from a narrow period of the species' breeding cycle in a disparate portion of the breeding distribution be critically reviewed and that local and regional information be incorporated prior to implementation, if management is deemed necessary at all, to avoid applying management prescriptions that may have the opposite of intended effects.

## ACKNOWLEDGEMENTS

We thank the J Refsnider lab, anonymous reviewers, S Graff, and an anonymous concerned reader for comments on the manuscript, and the originally published article. Our assessment was of the outcomes of management intended to benefit Golden-winged Warblers at Rice Lake NWR where we had data to evaluate a specific management action and neither the authors nor the University of Toledo, the University of Minnesota, the University of California, Berkeley, or any organization supporting our work intend to impugn the motives of organizations sponsoring or conducting the management we assessed. Rather, our intention is to use our opportunity to evaluate effects on a species of conservation concern from a specific management prescription that is being applied across the western Great Lakes region, and our hope is that management plans and programs can adapt their objectives and methods to better address high-priority conservation needs. Use of trade names does not imply endorsement by the US Geological Survey or the University of Minnesota. Survey data were collected as part of related studies of breeding Golden-winged Warbler ecology and migratory connectivity at the U.S. Geological Survey, Minnesota Cooperative Fish and Wildlife Research Unit.

### Funding

Data collection for this project was funded by the US Geological Survey, the US Fish and Wildlife Service, and the National Science Foundation. The funders had no role in study design, data collection and analysis, decision to publish, or preparation of the manuscript.

### Grant Disclosures

The following grant information was disclosed by the authors:
US Geological Survey.
US Fishand Wildlife Service.
National Science Foundation.

### Competing Interests

The authors declare there are no competing interests.

### Author Contributions

- Henry M. Streby conceived and designed the experiments, performed the experiments, analyzed the data, contributed reagents/materials/analysis tools, wrote the paper, prepared figures and/or tables.
- Gunnar R. Kramer performed the experiments, analyzed the data, wrote the paper, prepared figures and/or tables.
- Sean M. Peterson conceived and designed the experiments, performed the experiments, analyzed the data, wrote the paper, prepared figures and/or tables.
- David E. Andersen conceived and designed the experiments, contributed reagents/materials/analysis tools, wrote the paper.
## Animal Ethics

The following information was supplied relating to ethical approvals (i.e., approving body and any reference numbers):

Animals were captured and studied under two protocols approved by the University of Minnesota Institutional Animal Care and Use Committee.

## Field Study Permissions

The following information was supplied relating to field study approvals (i.e., approving body and any reference numbers):

Field research permits were issued by the USGS Bird Banding Laboratory (#21631) and the Minnesota Department of Natural Resources (19017).

## Data Availability

The raw data is provided in the Supplemental Files.

## Supplemental Information

Supplemental information for this article can be found online at http://dx.doi.org/10.7717/peerj.4319#supplemental-information.

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
