# Peer review of "Evaluating outcomes of young forest management on a target species of conservation concern"

_PeerJ, doi:10.7717/peerj.4319_

## Round 0.1 · original submission · Major Revisions

Following up with your appeal, I am issuing this “revised” decision so you can upload a point by point response to the reviewers, and an updated version of your manuscript.

· Appeal

Appeal

Dear Dr. Forbes:

Thank you and the Reviewers for your rapid and thorough handling of this manuscript. We are disappointed with the decision but we agree that Reviewer 1 has presented legitimate concerns. However, we believe we are mostly on the same page with Reviewer 1, and we are confident that the Reviewer's concerns can be reconciled if we are given the opportunity to respond and to revise the manuscript accordingly. We don't see any concerns from the reviewer that we could not address in a revision, including adding the parabolic analysis the Reviewer recommended. We therefore respectfully request from you the opportunity to respond and revise the manuscript.

Sincerely,

Henry Streby


· · Academic Editor

Reject

I found the concerns of Reviewer 1 to be highly compelling and thus the conclusions not supported by the data provided. With only two years of monitoring data post-management, it would seem that you do not have sufficient data to make the assertions that you do.

Reviewer 1 ·

Basic reporting

The Introduction describes to the reader a decline in Golden-winged Warbler population that has historically been attributed to loss of shrubland and increasing competition and hybridization with Blue-winged Warblers. Then, at the close of the first paragraph of the Introduction, the authors dismiss this decline in shrubland and instead point to non-breeding ground factors as the cause of the decline, asserting that ‘regional declines are more likely tied to non-breeding grounds factors’ and use as support the work of Kramer et al. (2017). Unfortunately, the argument is poorly made. Blue-winged Warblers now co-occur with Golden-winged Warblers in the Appalachians, which may entirely account for why newly created shrublands are not occupied by Golden-winged Warblers (i.e., they are out-competed regionally and locally, leaving considerable habitat unoccupied). Further, the assertion supported by Kramer et al. is determined by a total of 9 birds, 7 from Pennsylvania and 2 from Tennessee. This statement is too strongly stated, especially when Kramer et al. do not make this same assertion themselves. (It should be noted that three of the authors of the Kramer et al. paper author this current study as well.)

Line-specific comments......
Line 112. Warblers, not Warbles.

Lines 111-113. An alternative explanation is that the findings of the previous studies are usually correlative in nature, and it may very well be that the factors correlating to bird response are not most proximal to the species, but instead act as surrogates because they are factors we can readily measure. As such, the variation we see among studies may not be evidence of place-specific differences in bird response, but in place-specific differences in study methods.

Line 151. Replace ‘Ours’ with ‘Our study’.

Line 162. ‘our present our assessment’?

Line 219-221. This sentence is unclear. Did the adjacent ephemeral wetland not host Golden-winged Warblers prior or subsequent to vegetation management, or was it the managed area that did not host Golden-winged Warblers prior or subsequent to vegetation management? I suspect the latter, but cannot be sure.

Line 239 and line 369. ‘spatially explicit’. We never use a hyphen between an –ly adverb (‘spatially’), adjective (‘explicit’) combination modifying a noun (‘models’) or pronoun.

Line 253. Programming, not programing.

Line 259. Insert ‘at’ between ‘cover types’ and ‘predetermined’?

Line 261. This resolution is finer than the resolution of the ‘stand-level effects’ incorporated into the models.

Experimental design

The authors stack the deck against the management action by only examining a linear return to current conditions (e.g, lines 325-329, juveniles recruited 1.04 individuals after management linearly increasing to 1.45 individuals at the time conditions return to a pre-management state); it is likely, and expected, that during the return to a pre-management state that some period of improved performance will occur, over and above that which is observed currently. This improved performance may not be immediate (i.e., one or two years post-management action as in this study), but perhaps some period of years after. After this period of improved performance, the expectation is that this improved performance will decline back to the current state. If this response does not occur, it begs the question why one would implement this management action for Golden-winged Warblers in the first place, because all it can possibly do is exact harm upon the species. No one believes this to be the case, but more importantly this current study does not have the data to examine this issue. This is a debilitating concern that I do not believe the authors can correct if they want to comment on anything beyond the immediate period after management action.

Line-specific comments......
Lines 294-295. It strikes me that this assumption of a linear response by Golden-winged Warblers to management is faulty. The vegetation may respond linearly in its development or succession back to the pre-management state, but the idea behind why this management is expected to prove useful to Golden-winged Warblers is that the post-management state will at some point in time provide greater benefit than is currently observed. This benefit may not be immediate but, instead, may accumulate in time, and then fall off back to the currently observed response, suggesting a parabolic response rather than a linear one (for an example, see https://dfzljdn9uc3pi.cloudfront.net/2016/2156/1/fig-1-2x.jpg and think of the post-management state being on the far left and pre-management on the far right). If we allow only a linear response, then all we can achieve is a return to the pre-management level of biotic response, which begs the question as to why one would implement the conservation action if this is the expected result.

Validity of the findings

The conclusion that vegetation management failed to produce the desired result is reached after only two years of post-management monitoring of bird response. The authors themselves concluded (line 219) that it wasn’t until 3 years post-management that the managed area was visually similar to an adjacent unmanaged area.

Because of errors in assumptions, I have no confidence in the model results. There simply is no reason to believe a linear return to the current state would occur in bird response after management action. This assumption requires that the management action can only harm the species. This is logically unreasonable, and no literature is provided to support this notion.

Additional comments

I thought the manuscript was very nicely written, generally clear, and on a subject matter that should be interesting to a wide audience. Unfortunately, as currently crafted, I think the inferences are fatally flawed. The authors would be better served to restrict their analyses and inferences to the immediate period after management. The scenarios are simply unrealistic given their unsupported and illogical assumption of a linear return to conditions akin to the unmanaged state.

Reviewer 2 ·

Basic reporting

no comment

Experimental design

no comment

Validity of the findings

no comment

Additional comments

General Comments: I have very few issues with this manuscript. I think it is an important contribution to the GWWA conservation literature, and a timely assessment of management outcomes in the Western portion of the breeding range.

Specific Edits:
Line 51: Replace nest with net
Line 63: Remove “and purports to have benefited” – redundant
Line 87-88: Include reference to Fig1 to support the trends in GWWA decline
Line 89: Include citations that have referenced these “purported” causes of decline
Line 96: Including a rangemap (or at least reference to one) illustrating species distribution and density would be beneficial
Line 114-130: The impetus for the harvest effort described in this paragraph is unclear given a previous statement “Regional variation in Golden-winged Warbler population declines has likely led to this east-west density gradient, as Appalachian states have experienced declines of >95% since 1965 while Minnesota has seen stable or slightly increasing populations over the same period.” Please provide context by including a more explicit statement describing the need for this harvest/management initiative within the Western portion of the breeding range.
Line 131: Remove “on target species” – redundant
Line 138: Insert “However, the ABC initiative’s…”
Line 161 and 170: Contradiction in wording – one site and two sites. Consider rewording “We studied Golden-winged Warblers at 2 sites (one 80 ha managed site and one 30 ha control site)…”
Line 173-174: Redundant word choice. Consider using another word for “host”
Line 213-221: This section is dominated by results of management. Consider moving to first paragraph of results section.
Line 257: Replace "on" with "of"
Line 270 : Replace “of new” with “for”
Line 316-317: Retain information on core area shifts, but remove unnecessary behavioral description- “although some birds occasionally perched on sparse remaining shrubs or woody debris in the edges of managed areas”
Line 371: Increased area of grassland on the landscape not mentioned in results section. Consider moving post-harvest site description from Methods section to Results to clarify this transition. Further, include a more detailed description of the “grassland” habitat. I do not consider early successional habitat “dominated by a low (0.25 – 0.50 m), dense mat of dogwood” to be synonymous with grassland.
Line 454: Reword “is known not to do so”
Figure 2: Photo credit is unnecessary because photos were taken by authors
Figure 5: Unnecessary. These data are adequately described in the text.

---

## Round 0.2 · Minor Revisions

Dear Dr. Streby,

The more critical of the two original reviewers has evaluated your revised manuscript, and I am pleased to report that you have addressed the reviewer's major concerns successfully. The reviewer has a few remaining minor suggestions which I would ask you to include in your revised manuscript. If you can address these minor points, which I think you can easily do, I would be happy to accept your paper for publication.

Best regards,

Valery Forbes

Reviewer 1 ·

Basic reporting

No comment

Experimental design

No comment

Validity of the findings

No comment

Additional comments

I appreciate the efforts made by the authors to accommodate the concerns of the reviewers. I believe the authors' inclusion of an alternative scenario, one which paints the management action in a less negative light, only bolsters their argument. I agree with the authors' reasoning that abundance is likely linear in its relation to the amount of available habitat, at least at the end of the abundance spectrum they are witnessing in central Minnesota. Thus, the evaluation of a parabolic response solely on reproductive performance is a reasonable means of assessing alternative paths for population performance. Unlike the authors, though, I'm not convinced these alternatives are unrealistic. Do we have the data to suggest that they are? Not that I am aware. Perhaps these alternatives are unlikely, I do not know. I think it is reasonable to elide this connotation relating to its reasonable everywhere it is made, unless data exist to suggest otherwise.

Lines 446-448. I have pored over the Upper Mississippi River and Great Lakes Joint Venture documentation (i.e., The 2007 All-bird Implementation Plan, http://www.uppermissgreatlakesjv.org/docs/JV2007All-BirdPlanFinal2-11-08.pdf, and the 2007 Landbird Conservation Strategy, http://www.uppermissgreatlakesjv.org/docs/UMRGLR_JV_LandbirdHCS.pdf). I also did a literature search in Scopus and Google. I can find no where that the JV asserts 'Golden-winged Warbler needs are met.' This quote should be sourced. The link to the JV website is insufficient, as it does not lead the reader to the quote.

Line 451. 'where they are not replaced by Golden-winged Warblers'. I think you mean Blue-winged Warblers. Further, this revised sentence is a bit awkward in its construction. It begins as 'Even in areas where...' and then after the third comma it then says, again, 'even in areas where...'. I think this might be better crafted as two sentences.

---

## Round 0.3 · accepted · Accept

Thank you for your resubmission. Your responses to the remaining reviewer comments are satisfactory.